# Transverse Density Fluctuations around the Ground State Distribution of Counterions near One Charged Plate: Stochastic Density Functional View

**DOI:** 10.3390/e22010034

**Published:** 2019-12-25

**Authors:** Hiroshi Frusawa

**Affiliations:** Laboratory of Statistical Physics, Kochi University of Technology, Tosa-Yamada, Kochi 782-8502, Japan; frusawa.hiroshi@kochi-tech.ac.jp

**Keywords:** stochastic density functional theory, counterions, charged plate, strong coupling, the Wigner–Seitz cell, the Dean–Kawasaki equation

## Abstract

We consider the Dean–Kawasaki (DK) equation of overdamped Brownian particles that forms the basis of the stochastic density functional theory. Recently, the linearized DK equation has successfully reproduced the full Onsager theory of symmetric electrolyte conductivity. In this paper, the linear DK equation is applied to investigate density fluctuations around the ground state distribution of strongly coupled counterions near a charged plate, focusing especially on the transverse dynamics along the plate surface. Consequently, we find a crossover scale above which the transverse density dynamics appears frozen and below which diffusive behavior of counterions can be observed on the charged plate. The linear DK equation provides a characteristic length of the dynamical crossover that is similar to the Wigner–Seitz radius used in equilibrium theory for the 2D one-component plasma, which is our main result. Incidentally, general representations of longitudinal dynamics vertical to the plate further suggest the existence of advective and electrical reverse-flows; these effects remain to be quantitatively investigated.

## 1. Introduction

Water-soluble materials often have surface chemical groups that are dissociated in a polar solvent. Examples of such materials include not only mesoscopic particles, such as viruses, proteins, polyelectrolytes, membranes, and micelles, but also macroscopic objects like a glass plate of sample cell [1,2]. Both mesoscopic and macroscopic particles will be referred to here as ‘macroions’. The macroions are likely to carry a total surface charge exceeding thousands of elementary charges *e*, surrounded by oppositely charged counterions that are dissociated from the macroions [1,2]; counterions are electrostatically bound around macroions, due to the high asymmetry between counterions and macroions in valence of charges.

Focusing on the counterions, the macroion systems can be rephrased as inhomogeneous one-component ionic fluids in the presence of external fields—the one-component fluids of counterions can be regarded as the one-component plasma (OCP) [3,4,5]. Systems of charged particles immersed in a smooth neutralizing medium are commonly observed in nature, such as a suspension of dust grains in plasmas, as well as colloidal solutions, which can be modeled by the OCP in the unscreened limit of Yukawa fluids [3,4,5]. The 2D OCP has been used for the description of dusty plasmas confined by external fields, where the motion of particles interacting via 3D electrostatic interaction potential is restricted to a 2D surface [3,4,5]. There is, however, a crucial difference between counterion systems and the OCP, due to the localization of the electrically neutralizing background. While the whole space in the OCP is filled with a smooth background in either the 2D or 3D systems, counterions form a 3D electric double layer and are not neutralized unless they are localized on the macroion surfaces [6,7,8].

This paper will address the strong coupling systems of counterions in the presence of one charged plate, focusing especially on the transverse dynamics of density fluctuations around the ground state that will be specified in Section 2.1 [8,9,10,11]. Turning our attention to the dynamics, the strongly coupled counterion system is distinguished from the 2D OCP by one extra dimension, vertical to the macroion surface. Accordingly, density fluctuations occur not only along the 2D plane parallel to the macroion surface, but also along the one extra dimension. Furthermore, the strong coupling regime in the counterion systems may be realized at room temperature, and therefore dynamics due to thermal fluctuations need to be considered; however, there are few studies on the counterion dynamics in the ground state.

Thus, the main aim of this paper is to investigate the anisotropic fluctuation field n(r,t) of counterion density due to the coarse-grained dynamics of counterion density $\rho \left(r,t\right)=\rho_{\infty }\left(r\right)+n\left(r,t\right)$ around a ground state distribution $\rho_{\infty }\left(r\right)$, using the stochastic density functional equation. The stochastic density functional theory is based on the so-called Dean–Kawasaki (DK) equation that describes the evolution of the instantaneous microscopic density field of overdamped Brownian particles [12,13,14,15,16,17,18,19,20]. The stochastic density functional theory has been used as one of the most powerful tools for describing slowly fluctuating and/or intermittent phenomena [16,17,18,19,20], such as glassy dynamics, nucleation or pattern formation of colloidal particles, stochastic thermodynamics of colloidal suspensions, dielectric relaxation of Brownian dipoles, and even tumor growth.

The original DK equation includes nonlinear terms of dynamic origin due to the kinetic coefficient that is proportional to fluctuating field ρ(r,t) [12,13,14,15,16,17,18,19,20]. While the nonlinearility of the original DK equation leads to the above successful descriptions of various phenomena [16,17,18,19,20], a more tractable form is required. It has been recently demonstrated that the DK equation can be linearized with respect to n(r,t) when $n\left(r,t\right)/\rho_{\infty }\left(r\right)\ll 1$, and that the linear stochastic equation of density fluctuations is of great practical use [21,22,23,24,25,26,27]. The density fluctuations of fluids near equilibrium are surprisingly well described by model-B dynamics of a Gaussian field theory whose effective quadratic Hamiltonian for the density fluctuation field is constructed to yield the exact form of the static density-density correlation function [25]. Furthermore, we have demonstrated that the DK equation can be directly linearized in the first approximation of the driving force due to the free energy functional F[ρ] of an instantaneous density distribution ρ, when small density fluctuations around a metastable state are considered [21].

The stochastic thermodynamics around a metastable state has been investigated using the stochastic density functional equation (the DK equation), showing that the heat dissipated into the reservoir is generally negligible [21]. The linear stochastic density functional theory has also been found relevant to investigate out-of-equilibrium phenomena, including the formulations of the full Onsager theory of electrolyte conductivity [22].

The remainder of this paper is organized as follows: Section 2 provides formal background in the case of a single charged plate system. We give the linear DK equation as a stochastic density functional equation, after specifying a general form of the free energy functional F[ρ] of a given density. In Section 3, the linear DK equation is applied to the strongly coupled counterion system by considering density fluctuations $n=\rho -\rho_{\infty }$ around $\rho_{\infty }$. We can verify that the first derivative of F[ρ] in the ground state (i.e., ${\delta F\left[\rho \right]/\delta \rho |}_{\rho =\rho_{\infty }}$) produces a constant, similar to the above metastable state. Accordingly, the DK equation of the counterion system can be linearized around the ground state. We will also see the underlying physics of anisotropic fluctuations (vertical to the plate) in terms of the general form of the linear DK equation. In Section 4, we focus on the transverse dynamics along the plate surface, assuming the absence of the gradient of the fluctuating density field vertical to the plate. First, we derive the frozen dynamics over a long-range scale beyond the Wigner–Seitz cell, reflecting the formation of the Wigner crystal on the charged plate. Furthermore, the linear DK equation determines a crossover length $l_{c}$, below which we can observe diffusive behaviors of counterions condensed on the plate. Our main result in this study is the quantitative evaluation of $l_{c}$, yielding $l_{c}∼a$ for $\Xi ∼10^{3}$. Section 5 contains a summary and conclusions.

## 2. Formal Background

### 2.1. Ground State of Counterion System in the Strong Coupling Limit

Let us briefly summarize what has been achieved by theoretical and simulation studies on the OCP and the counterion systems in the strong coupling regime.

The thermodynamics of the OCP system is characterized by the coupling parameter [3,4,5],
(1)$$\Gamma =\frac{q^{2}l_{B}}{a},$$
where *q* is the valence of counterions, $l_{B}\equiv e^{2}/4\pi \epsilon k_{B}T$ is the distance (the so-called Bjerrum length) at which two elementary charges interact electrostatically with thermal energy $k_{B}T$, when they are surrounded by a polar solvent with its dielectric permittivity and temperature being ϵ and *T*, and the Wigner–Seitz cell radius *a* defined by $(\pi a^{2})\sigma =q$ using the surface number density σ of macroion charges [3,4,5,7]. Thermodynamic properties of OCP systems have been extensively studied over decades and accurate numerical results as well as their fits are available in the literature [3,4,5]. As Γ increases, the OCP shows a transition from a weakly coupled gaseous regime (Γ≪1) to a strongly coupled fluid regime (Γ≫1), and it eventually crystallizes. The concept of the Wigner crystallization due to long-range electrostatic interactions underlies the formation of colloidal crystals, or photonic crystals with large lattice constant, comparable in magnitude to the wavelength of visible light [2,3,4,5].

Meanwhile, in counterion systems, a Wigner–Seitz radius *a* has not been adopted in rescaling the Bjerrum length as $l_{B}/a$. We have used another coupling parameter Ξ defined by [6,7,8,9,10,11]
(2)$$\begin{array}{cc} \; & \Xi =\frac{q^{2}l_{B}}{\lambda }, \end{array}$$
(3)$$\begin{array}{cc} \; & \lambda =\frac{1}{2\pi ql_{B}\sigma }, \end{array}$$
using the Gouy–Chapman length $\lambda =1/(2\pi ql_{B}\sigma )$, a characteristic length of the electric double layer. Inserting the relation $\sigma =q/(\pi a^{2})$ into the definition of λ, we have
(4)$$\frac{a}{\lambda }=2\pi q\sigma l_{B}a=\frac{2q^{2}l_{B}}{a}=2\Gamma .$$
It follows from Equations (3) and (Equation 4) that
(5)$$\Xi =2\Gamma^{2}.$$
When macroion-counterion attractions are weak, the structure of such ionic cloud, characterized by λ, will be dispersed instead of forming the 2D OCP. The dispersed electric double layer is thus represented by the weak coupling parameter of Ξ≪1, where the Poisson–Boltzmann approach and its systematic improvements via the loop expansion has been found relevant [6,7,8]. On the other hand, as Ξ increases while reducing λ, the electric double layer thins and the coarse-grained distribution of counterions becomes two-dimensional [6,7,8,9,10,11]. Correspondingly, *a* becomes much larger than λ in the strong coupling regime of $\Xi =2\Gamma^{2}\gg 1$ (i.e., a≫λ), as found from Equations (Equation 4) and (Equation 5).

A field theoretical treatment provides counterion density distribution, $\rho_{\infty }\left(z_{0}\right)$, in the ground state of the strong coupling limit (Ξ,Γ→∞) [6,8]:(6)$$\begin{array}{c} \rho_{\infty }\left(z_{0}\right)=\frac{\sigma }{\lambda }\exp\;\left\{-J\left(r_{0}\right)\;\right\}, \end{array}$$
where $J(r_{0})$ denotes the external electrostatic potential in the $k_{B}T$-unit due to macroion-counterion interactions. In a single charged plate system, $J(r_{0})$ is expressed as $J\left(r_{0}\right)=z_{0}/\lambda$ with $z_{0}$ denoting the distance between the position $r_{0}$ and the charged plate; therefore, Equation (Equation 6) implies that a large portion of the counterions are condensed within the thin electric double layer, which supports the observation that strongly coupled counterions behave like the 2D OCP. Extensive Monte Carlo simulations have been performed on the strong coupling regimes of counterions, especially for one- and two-plate systems [8,9,10,11]. Accordingly, the asymptotic behavior given by Equation (Equation 6) has been corroborated by simulation results on the counterion distributions. The correction to Equation (Equation 6) has also been evaluated in detail, based on the simulation results. For about two decades, a variety of strong coupling theories have been developed to explain the above simulation results in the strong coupling regime, focusing not only on the validation of the longitudinal distribution mimicked by the ideal gas behavior (i.e., Equation (Equation 6)) in the vicinity of the charged plate, but also on the deviations from Equation (Equation 6) for $z_{0}>\lambda$; see [8] for a recent review.

### 2.2. Imposing a Given Density Distribution ρ on the Grand Potential Ω

Let $\hat{\rho }\left(r\right)$ be an instantaneous density of counterions located at $r_{i}\;\left(i=1,\cdots ,N\right)$, where the counterion system is rescaled as $r=\left(x,y,z\right)=\left(x_{0}/a,y_{0}/a,z_{0}/a\right)=r_{0}/a$. Figure 1 shows a schematic of the rescaled system. The instantaneous density is expressed as
(7)$$\hat{\rho }\left(r\right)=a^{3}\sum_{i=1}^{N}\delta \left(r-r_{i}\right),$$
the use of which macroion-counterion interaction energy for a one-plate system transforms the configurational representation, given by Equation (Equation 51) of Appendix B, to
(8)$$\begin{array}{cc} \Delta U_{cm}\left\{\hat{\rho }\right\} & =\int drJ\left(r\right)\hat{\rho }\left(r\right),\\ J(r) & =2\Gamma z, \end{array}$$
which is a functional of $\hat{\rho }$ (see Appendix B for the details).

We can impose a given density distribution ρ(r) on the counterion system via the following delta functional [14,19,20,21]:(9)$$\prod_{r}\delta \left[\hat{\rho }\left(r\right)-\rho \left(r\right)\right]=\int D\psi \;e^{\int dr\;\;i\psi \left(r\right)\{\hat{\rho }\left(r\right)-\rho \left(r\right)\}},$$
where the potential field ψ(r) has been introduced in the Fourier transform of the delta functional. Multiplying the configurational integral representation of the grand potential Ω[J] (see Appendix B for the definition) by the constraint in Equation (Equation 9), the formal expression of F[ρ] is obtained:(10)$$\begin{array}{cc} e^{-F[\rho ]}\; & =\prod_{r}\delta \left[\hat{\rho }\left(r\right)-\rho \left(r\right)\right]e^{-\Omega [J]}\\ \; & =e^{-\int drJ(r)\rho (r)}\int D\psi \;e^{-\Omega [-i\psi ]-\int dr\;\;i\psi (r)\rho (r)}, \end{array}$$
where J(r) given in Equation (Equation 8) represents the external potential created by the charged plate. In the mean-field approximation, we obtain (see also Appendix C)
(11)$$\begin{array}{cc} F[\rho ] & =A[\rho ]+\int drJ(r)\rho (r), \end{array}$$
(12)$$\begin{array}{cc} A[\rho ] & =\Omega \left[\psi^{*}\right]-\int dr\psi^{*}\left(r\right)\rho \left(r\right), \end{array}$$
where the saddle-point potential field $\psi^{*}$ satisfies the following relation:(13)$${\left.\frac{\delta \left(\beta \Omega [-i\psi ]\right)}{\delta \psi (r)}\right|}_{\psi =i\psi^{*}}=-i\rho \left(r\right).$$
The first Legendre transform of $\Omega [\psi^{*}]$ provides the Hohenberg–Kohn free energy A[ρ] defined by Equation (12), the central functional of the equilibrium density functional theory [28,29].

### 2.3. Stochastic Density Dynamics Obeying the Dean–Kawasaki Equation

Here we focus on the stochastic dynamics of a density field at time *t*, ρ(r,t), whose spatially varying distribution is the same as the coarse-grained variation of $\hat{\rho }\left(r\right)$. What matters in terms of the stochastic density dynamics is the free energy functional F[ρ] of a given density field ρ, rather than the grand potential Ω in equilibrium. For the density functional, we have provided the approximate form of F[ρ]. The driving force due to F[ρ] and the density-dependent multiplicative noise $\zeta [\rho ,\vec{\eta }]$ creates the stochastic dynamics that obeys the DK equation [13,14,15,16,17,18,19,20]:(14)$$\partial_{t}\rho =\nabla \cdot D\rho \;\nabla \frac{\delta F[\rho ]}{\delta \rho }+\zeta \left[\rho ,\vec{\eta }\right],$$
where we have introduced a scaled diffusion constant, $D=D_{0}/a^{2}$, using the bare diffusion constant $D_{0}$ and the spatio-temporal average of the multiplicative noise correlations is given by
(15)$$\begin{array}{c} \langle \;\zeta \left[\rho \left(r,t\right),\vec{\eta }\left(r,t\right)\right]\;\zeta \left[\rho \left(r^{\prime },t^{\prime }\right),\vec{\eta }\left(r^{\prime },t^{\prime }\right)\right]\;\rangle =-2D\delta \left(t-t^{\prime }\right)\nabla_{r}\cdot \rho \left(r,t\right)\nabla_{r}\delta \left(r-r^{\prime }\right), \end{array}$$
with the vectorial white noise field $\vec{\eta }\left(r,t\right)$ that has the correlation $\langle \eta_{l}\left(r,t\right)\eta_{m}\left(r^{\prime },t^{\prime }\right)\rangle =\delta_{lm}\delta \left(r-r^{\prime }\right)\delta \left(t-t^{\prime }\right)$. Equation (Equation 15) can read [13,14,15,16,17,18,19,20,21,22,23,24,25,26,27]
(16)$$\zeta \left[\rho ,\eta \right]=\nabla \cdot \sqrt{2D\rho (r,t)}\vec{\eta }\left(r,t\right).$$
In general, the stochastic Equation (Equation 14) includes not only the multiplicative noise term, but also the nonlinear term associated with F[ρ]. As shown below, however, a linear DK equation may be used to investigate the stochastic density dynamics due to fluctuations of counterions in the ground state by expanding Equation (Equation 14) around $\rho_{\infty }$.

## 3. Stochastic Density Functional Equation for Fluctuations round the Ground State Distribution $\rho_{\infty }$

### 3.1. Linearizing the Stochastic Dean–Kawasaki Equation (Equation 14)

Expanding the first derivative of F[ρ] around $\rho_{\infty }$, we have
(17)$$\begin{array}{cc} \nabla \frac{\delta F[\rho ]}{\delta \rho (r,t)}\; & =\nabla \left[{\left.\frac{\delta F[\rho ]}{\delta \rho (r,t)}\right|}_{\rho =\rho_{\infty }}+\int dr^{\prime }{\left.\frac{\delta^{2}F\left[\rho \right]}{\delta \rho \left(r,t\right)\delta \rho \left(r^{\prime },t\right)}\right|}_{\rho =\rho_{\infty }}n\left(r^{\prime }\right)\right]\\ \; & =\nabla \left[-\int dr^{\prime }c\left(r-r^{\prime }\right)n\left(r^{\prime },t\right)+\frac{n(r,t)}{\rho_{\infty }\left(z\right)}\right], \end{array}$$
(18)$$\begin{array}{c} \rho \left(r,t\right)=\rho_{\infty }\left(r\right)+n\left(r,t\right), \end{array}$$
due to
(19)$$\nabla \cdot D\rho \left(r\right){\left.\nabla \frac{F[\rho ]}{\delta \rho }\right|}_{\rho =\rho_{\infty }}=0;$$
See Appendix D for the details. It follows from Equations (Equation 17) and (Equation 19) that the right-hand side (rhs) of Equation (Equation 14) reads
(20)$$\begin{array}{cc} \nabla \cdot D\rho \nabla \frac{\delta F[\rho ]}{\delta \rho (r,t)}+\zeta \left[\rho ,\vec{\eta }\right]\; & =\nabla \cdot D\rho_{\infty }\left(1+\frac{n(r,t)}{\rho_{\infty }}\right)\nabla \left\{-\int dr^{\prime }c\left(r-r^{\prime }\right)n\left(r^{\prime },t\right)+\frac{n(r,t)}{\rho_{\infty }\left(z\right)}\right\}+\zeta \left[\rho_{\infty }\left(1+\frac{n(r,t)}{\rho_{\infty }}\right),\vec{\eta }\right]\\ \; & \approx \nabla \cdot D\rho_{\infty }\nabla \left\{-\int dr^{\prime }c\left(r-r^{\prime }\right)n\left(r^{\prime },t\right)+\frac{n(r,t)}{\rho_{\infty }\left(z\right)}\right\}+\zeta \left[\rho_{\infty },\vec{\eta }\right], \end{array}$$
when $n/\rho_{\infty }\ll 1$. It is to be noted that the ideal gas distribution $\rho_{\infty }\left(r\right)$ given by Equation (Equation 6) reproduces only the longitudinal distribution of simulation results in the vicinity of a highly charged plate, resulting from attractive and repulsive Coulomb interactions in the strong coupling limit (see also Appendix A). Correspondingly, Equation (Equation 20) reveals that the dynamics of fluctuating density n(r,t) is governed by the strong Coulomb interactions as represented by the contribution, $-\int dr^{\prime }c\left(r-r^{\prime }\right)n\left(r^{\prime },t\right)$, on the rhs of Equation (Equation 20). In Equations (Equation 17) and (Equation 20), the direct correlation function $c(r-r^{\prime })$ appears because F[ρ] is expressed using the Hohenberg–Kohn free energy functional A[ρ] as given by Equation (12) [28,29]. We also have
(21)$$\partial_{t}\left\{\rho_{\infty }\left(z\right)+n\left(r,t\right)\right\}=\partial_{t}n\left(r,t\right),$$
due to $\partial_{t}\rho_{\infty }=0$, on the left hand side of the DK Equation (Equation 14).

Combining Equations (Equation 14), (Equation 20), and (Equation 21), we obtain the linear DK equation:(22)$$\begin{array}{cc} \partial_{t}n\left(r,t\right) & =D\nabla^{2}n\left(r,t\right)+D\rho_{\infty }\left(z\right)\nabla^{2}\psi_{n}\left(r,t\right)-\nabla \cdot j_{⊥}\left(r,t\right)+\zeta \left[\rho_{\infty },\vec{\eta }\right], \end{array}$$
where $\psi_{n}$ denotes a fluctuating Coulomb potential defined by
(23)$$\psi_{n}\left(r,t\right)\equiv -\int dr^{\prime }c\left(r-r^{\prime }\right)n\left(r^{\prime }\right),$$
and the longitudinal current $j_{⊥}\left(r,t\right)=\left(0,\;0,\;j_{z}\right)$, which is along the *z*-axis vertical to the charged plate, arises from the gradient of the ground density distribution $\nabla \rho_{\infty }$. Incidentally, in the rescaled system of $\rho_{\infty }\left(r,t\right)=a^{3}\rho_{\infty }\left(r_{0},t\right)$, Equation (Equation 6) is rewritten as
(24)$$\rho_{\infty }\left(z\right)=\frac{\sigma a^{3}}{\lambda }e^{-2\Gamma z},$$
thereby providing
(25)$$\nabla \rho_{\infty }=\left(0,\;0,\;-2\Gamma \rho_{\infty }\right),$$
in the rescaled system. In the next subsection, we will discuss the details of $j_{⊥}\left(r,t\right)$ associated with the presence of $\nabla \rho_{\infty }$ in the longitudinal direction.

Equation (Equation 22) appears to be a simple extension of the Poisson–Nernst–Planck equation [30] when the longitudinal current $j_{⊥}\left(r,t\right)$ disappears. However, in actuality, the Poisson-like equation in the second term on the rhs of Equation (Equation 22) differs from the conventional Poisson equation because the interaction potential is replaced by the direct correlation function. In this paper, we adopt the direct correlation function, being of the following form [31,32,33]: (26)$$\begin{array}{cc} -c(r) & =\int dr^{\prime }\frac{\Gamma }{|r-r^{\prime }|}g_{a}\left(r^{\prime }\right)=\Gamma \;{\tilde{v}}_{L}\left(r\right), \end{array}$$
(27)$$\begin{array}{cc} g_{a}\left(r-r^{\prime }\right) & =\frac{e^{-|r-r^{\prime }{|}^{2}/m}}{{\left(m\pi \right)}^{3/2}}=\frac{e^{-|r_{0}-r_{0}^{\prime }{|}^{2}/\left(ma^{2}\right)}}{{\left(ma^{2}\pi \right)}^{3/2}}\;\left(m=1.08^{-2}\right), \end{array}$$
(28)$$\begin{array}{cc} {\tilde{v}}_{L}\left(r\right) & =\frac{\mathrm{erf}(1.08r)}{r}\;(r\equiv |r|=|r_{0}\left|/a\right), \end{array}$$
which gives
(29)$$\begin{array}{cc} \nabla^{2}\psi_{n}\left(r,t\right) & \equiv -\int dr^{\prime }\nabla^{2}c\left(r-r^{\prime }\right)n\left(r^{\prime },t\right)\\ \; & =-4\pi \Gamma \int dr^{\prime }g_{a}\left(r-r^{\prime }\right)n\left(r^{\prime },t\right)\\ \; & =-4\pi \Gamma \tilde{n}\left(r,t\right), \end{array}$$
where $\tilde{n}\left(r,t\right)$ denotes a coarse-grained density that is smeared by the Gaussian distribution function $g_{a}\left(r-r^{\prime }\right)$ over a range of the Wigner–Seitz radius *a*. The above form of the direct correlation function has been demonstrated to be available for the OCP in the strong coupling regime of Γ≫1. In Equation (Equation 26), the bare electrostatic potential (∼1/r) is modified using the Gaussian distribution function $g_{a}\left(r\right)$, and the second equation of Equation (Equation 26) introduces the function of ${\tilde{v}}_{L}\left(r\right)=\mathrm{erf}\left(1.08r\right)/r$ that represents the long-range part of the Coulomb interaction potential [31,32]. It is to be noted that the internal energy of the OCP, obtained using this direct correlation function, or Equation (Equation 26), exhibits an error of less than 0.8% in the strong coupling regime [31].

Considering that the Fourier transform ${\tilde{v}}_{L}\left(k\right)$ of ${\tilde{v}}_{L}\left(r_{0}/a\right)$ is given by
(30)$${\tilde{v}}_{L}\left(k\right)=\frac{4\pi }{k^{2}}e^{-k^{2}\left(ma^{2}\right)/4},$$
Equation (Equation 29) is rewritten in the original coordinate of $r_{0}$ as
(31)$$\begin{array}{cc} \nabla^{2}\psi_{n}\left(r_{0},t\right) & =-k^{2}\psi_{n}\left(k,t\right)\\ \; & =-4\pi q^{2}l_{B}e^{-k^{2}\left(ma^{2}\right)/4}\;n\left(k,t\right), \end{array}$$
using the Fourier transforms of $\psi_{n}\left(r_{0},t\right)$ and $n(r_{0},t)$: $\psi_{n}\left(k,t\right)$ and n(k,t). In the limit of a→0, Equation (Equation 29) is reduced to the conventional Poisson equation. It follows from Equations (Equation 29) and (Equation 31) that the Fourier transform $\tilde{n}\left(k,t\right)$ of the coarse-grained density $\tilde{n}\left(r_{0},t\right)$ reads
(32)$$\begin{array}{c} \tilde{n}\left(k,t\right)=e^{-k^{2}\left(ma^{2}\right)/4}\;n\left(k,t\right), \end{array}$$
implying the cut-off of $\tilde{n}\left(k,t\right)$ at a high wavenumber in correspondence with the coarse-graining of n(r,t).

### 3.2. Implications of Longitudinal Contributions Given by Equation (Equation 33)

As described in Section 3.1, the gradient of $\rho_{\infty }\left(r\right)$ has only the *z*-component as found from Equation (Equation 25). The *z*-component given by $\partial_{z}\rho_{\infty }=-2\Gamma \rho_{\infty }$ yields the longitudinal contribution, $-\nabla \cdot j_{⊥}\left(r,t\right)$, to the rhs of Equation (Equation 22):(33)$$\begin{array}{cc} -\nabla \cdot j_{⊥}\left(r,t\right) & =-\partial_{z}j_{z}\left(r,t\right)\\ \; & =2\Gamma D\partial_{z}n\left(r,t\right)-D\partial_{z}\rho_{\infty }\left(z\right)\Gamma E_{z}\left(r,t\right)\\ \; & =2\Gamma D\left\{\partial_{z}n\left(r,t\right)+\rho_{\infty }\left(z\right)\Gamma E_{z}\left(r,t\right)\right\}, \end{array}$$
where $\Gamma E_{z}\left(r,t\right)$ denotes the *z*-component of fluctuating electric field E(r,t) defined by
(34)$$E\left(r,t\right)=-\nabla \psi_{n}\left(r,t\right)=\Gamma \left[\begin{array}{c} E_{x}\left(r,t\right)\\ E_{y}\left(r,t\right)\\ E_{z}\left(r,t\right) \end{array}\right].$$

It is found from Equations (Equation 23), (Equation 33), and (Equation 34) that $-\nabla \cdot j_{⊥}\left(r,t\right)$ disappears in the absence of the longitudinal variance in n(r,t) (i.e., $\partial_{z}n\left(r,t\right)=0$) as well as $\rho_{\infty }\left(r\right)$, which is the reason why $j_{⊥}\left(r,t\right)$ has been referred to as the longitudinal current. The two terms on the rhs of Equation (Equation 33) are expressed in the original coordinate as follows: (35)$$\begin{array}{cc} \; & 2\Gamma D\partial_{z}n\left(r,t\right)=\frac{a}{\lambda }\left(\frac{D_{0}}{a^{2}}\right)\partial_{z}n\left(r,t\right)=\frac{D_{0}}{\lambda }\partial_{z_{0}}n\left(r_{0},t\right), \end{array}$$
(36)$$\begin{array}{cc} \; & 2\Gamma D\rho_{\infty }\left(z\right)\Gamma E_{z}\left(r,t\right)=\frac{a}{\lambda }\left(\frac{D_{0}}{a^{2}}\right)\rho_{\infty }\left(z\right)q^{2}l_{B}E_{z}\left(r_{0},t\right)=D_{0}\rho_{\infty }\left(z\right)\left\{\frac{q^{2}l_{B}E_{z_{0}}\left(r_{0},t\right)}{\lambda }\right\}. \end{array}$$
Equation (Equation 35) represents a vertical advection of fluctuating density field $n(r_{0},t)$. Putting this advection term given by Equation (Equation 35) on the left hand side of Equation (Equation 22), the combination of Equations (Equation 33)–(Equation 36) transforms Equation (Equation 22) to
(37)$$\begin{array}{c} \partial_{t}n\left(r_{0},t\right)\underline{-\frac{D_{0}}{\lambda }\partial_{z_{0}}n\left(r_{0},t\right)}=D_{0}\nabla^{2}n\left(r_{0},t\right)-D_{0}q^{2}l_{B}\rho_{\infty }\left(r_{0},t\right)\left\{4\pi \tilde{n}\left(r_{0},t\right)\underline{-\frac{E_{z_{0}}\left(r_{0},t\right)}{\lambda }}\right\}+\zeta \left[\rho_{\infty },\vec{\eta }\right] \end{array}$$
in the original coordinate representation, where the underlined terms corresponding to the longitudinal contributions. The former contribution, the second term on the left hand side of Equation (Equation 37), suppresses density fluctuations, whereas the latter, the third term on the rhs of Equation (Equation 37), acts as a positive feedback to enhance counterion condensation in proximity to the charged plate. Figure 2 is a schematic of such opposite roles of longitudinal contributions from the above underlined terms.

On the one hand, the underlined term on the left hand side of Equation (Equation 37) represents the advective flow term. The advection velocity is given by $D_{0}/\lambda$, which increases as the Gouy–Chapman length λ, a characteristic length of the electric double layer, is shorter. The negative sign of this term indicates that the flow direction is always in the opposite direction to the *z*-axis. Figure 2 illustrates translation of whole fluctuating density field n(r,t), like a Goldstone-mode, due to the advection flow. Figure 2 shows the case of $\partial_{z_{0}}n<0$ where the increase from $\rho_{\infty }\left(0\right)$ (i.e., $n(r_{0},t)>0$ at $z_{0}=0$) is lowered when $\partial_{z_{0}}n<0$, and vice versa because of the fixed direction of the advection flow. In other words, density fluctuations are suppressed due to the former longitudinal contribution associated with advection flow.

On the other hand, the underlined contribution of the third term on the rhs of Equation (Equation 37) arises from the $z_{0}$-component $\Gamma E_{z_{0}}$ of a fluctuating electric field $E(r_{0},t)$; however, this term reduces the third term on the rhs of Equation (Equation 37), or the smeared density $\tilde{n}\left(r_{0},t\right)$ induced by $E(r_{0},t)$ itself. We should remember that, in the limit of a→0, the second term on the rhs of Equation (Equation 37) corresponds to the electrostatic term that is associated with the Poisson equation: we have $-D_{0}q^{2}l_{B}\rho_{\infty }\left(r_{0},t\right)4\pi n\left(r_{0},t\right)$ with $\tilde{n}\left(r_{0},t\right)$ replaced by the original density $n(r_{0},t)$, indicating the electrostatically restoring term to $n(r_{0},t)\to 0$ as represented by the negative sign. Accordingly, the latter longitudinal contribution plays a role of positive feedback for counterion condensation, as opposed to the above electrostatic suppression. In actuality, the latter term increases the strength of longitudinal fluctuating field when $E_{z_{0}}>0$, or $\partial_{z_{0}}n\left(r_{0},t\right)<0$, which means that the counterions have become more condensed. Meanwhile, the negative sign of $E_{z_{0}}<0$ yields the restoring contribution to $n(r_{0},t)\to 0$ when accumulated counterions leave the charged plate: $\partial_{z_{0}}n\left(r_{0},t\right)>0$. In both cases of $E_{z_{0}}>0$ and $E_{z_{0}}<0$, the latter longitudinal enhances counterion condensation, which is represented as electrical reverse-flow for $\partial_{z_{0}}n\left(r_{0},t\right)<0$ in Figure 2.

## 4. Density-Density Correlations Due to Transverse Dynamics Along the Plate Surface

Supposing that $\partial_{z_{0}}n\left(r_{0},t\right)=0$, we can focus on the transverse dynamics parallel to the charged plate at $z_{0}=0$, which we will investigate quantitatively. With the use of the Fourier transform n(k,t) of $n(r_{0},t)$, Equation (Equation 37) becomes
(38)$$\begin{array}{cc} \partial_{t}n\left(k,t\right)\; & =-D_{0}\left\{k^{2}+4\pi q^{2}l_{B}\rho_{\infty }\left(0\right)e^{-k^{2}\left(ma^{2}\right)/4}\right\}n\left(k,t\right)+\zeta \left[\rho_{\infty },\vec{\eta }\right]\\ \; & =-D_{0}\Xi \;G\left(k\right)n\left(k,t\right)+\zeta \left[\rho_{\infty },\vec{\eta }\right], \end{array}$$
(39)$$\begin{array}{cc} G(k) & =\frac{k^{2}}{\Xi }+4\pi \sigma e^{-k^{2}\left(ma^{2}\right)/4}, \end{array}$$
given that $\partial_{z_{0}}n\left(r_{0},t\right)=0$. In Equation (Equation 39), the conventional coupling constant $\Xi =q^{2}l_{B}/\lambda$ given by Equation (Equation 2) appears using $\rho_{\infty }\left(0\right)=\sigma /\lambda$. In both of the strong coupling regime (Ξ≫1) and the low wavenumber region (ka≪1), the propagator G(k) is approximated by
(40)$$\begin{array}{c} G\left(k\right)\approx \frac{1}{a^{2}}\left\{\frac{{\left(ka\right)}^{2}}{\Xi }+4q\right\}\approx \frac{4q}{a^{2}}=4\pi \sigma , \end{array}$$
using the relation $\sigma =q/(\pi a^{2})$.

Now we have the analytical solution to Equation (Equation 38) in the following form [25]:(41)$$\begin{array}{cc} n(r_{0},t) & =e^{-tD_{0}\Xi \;G\left(r_{0}\right)}n\left(r_{0},0\right)+\int_{0}^{t}dse^{-\left(t-s\right)D_{0}\Xi \;G\left(r_{0}\right)}\zeta \left[\rho_{\infty }\left(z_{0}\right),\eta \left(r_{0},s\right)\right]. \end{array}$$
Considering the real space representation that $D_{0}\Xi G\left(r_{0}\right)\approx 4\pi D_{0}\Xi \sigma$ in the above approximation of Equation (Equation 40), the above exponential factors, $e^{-tD_{0}\Xi \;G}$ and $e^{-\left(t-s\right)D_{0}\Xi \;G}\;\left(s<t\right)$, are negligible due to Ξ≫1. Hence, Equation (Equation 41) is reduced to $n\left(r_{0},t\right)=\zeta \left[\rho_{\infty }\left(z_{0}\right),\eta \left(r_{0},s\right)\right]$, thereby providing
(42)$$\begin{array}{cc} \langle n\left(r_{0},t\right)n\left(r_{0}^{\prime },t^{\prime }\right)\rangle & =\langle \zeta \left[\rho_{\infty }\left(z_{0}\right),\vec{\eta }\left(r_{0},t\right)\right]\zeta \left[\rho_{\infty }\left(r_{0}^{\prime }\right),\vec{\eta }\left(r_{0}^{\prime },t^{\prime }\right)\right]\rangle \\ \; & =2D_{0}\rho_{\infty }\left(z_{0}\right)\delta \left(r_{0}-r_{0}^{\prime }\right)\delta \left(t-t^{\prime }\right). \end{array}$$
It is found from Equation (Equation 42) that there is no correlation of transverse density fluctuations, which represents a coarse-grained frozen dynamics in the strong coupling regime of coarse-grained 2D OCP.

We can determine a crossover scale $l_{c}$. or associated crossover wavenumber $k_{c}=2\pi /l_{c}$ by comparing two terms on the rhs of Equation (Equation 39). In the above approximation, the first term on the rhs of Equation (Equation 39) has been neglected based on the condition that Ξ≫1 and ka≪1. While increasing the wavenumber and maintaining the strong coupling of Ξ≫1, we arrive at the crossover wavenumber $k_{c}$ that is defined by the following relation:(43)$$\begin{array}{c} \frac{k_{c}^{2}}{\Xi }=4\pi \sigma e^{-k_{c}^{2}\left(ma^{2}\right)/4}, \end{array}$$
stating that the two terms on the rhs of Equation (Equation 39) are comparable to each other. We now introduce the main branch $W_{0}\left(x\right)$ of the Lambert *W*-function [34], so that Equation (Equation 43) is converted to
(44)$$\begin{array}{cc} \nu & =W_{0}\left(\nu e^{\nu }\right)=W_{0}\left(qm\Xi \right),\\ \nu & \equiv k_{c}^{2}\left(ma^{2}\right)/4, \end{array}$$
based on another expression of Equation (Equation 43) as follows:(45)$$\begin{array}{c} \nu e^{\nu }=\pi \sigma \Xi \left(ma^{2}\right)=qm\Xi . \end{array}$$

The approximate form of $W_{0}\left(x\right)\approx \ln x$ for x≫1 [34] applies to Equation (Equation 44) because of qmΞ≫1. It follows that Equation (Equation 44) reads
(46)$$\begin{array}{c} 2\pi \left(\frac{a}{l_{c}}\right)=k_{c}a\approx \frac{2}{m^{1/2}}\sqrt{\ln(qm\Xi )}=2.16\sqrt{\ln(qm\Xi )}, \end{array}$$
which is our main result in this study. Below this scale specified by the crossover length $l_{c}$, we can observe the diffusive behavior of counterions, instead of frozen correlations represented by Equation (Equation 42). Taking $qm\Xi =10^{4}$ (or $\Xi ∼10^{3}$ for q∼10) as an example of the strong coupling regime, Equation (Equation 46) provides
(47)$$\begin{array}{cc} k_{c}a & \approx 6.56,\\ \frac{l_{c}}{a} & \approx \frac{2\pi }{6.56}. \end{array}$$
The latter relation implies that the transverse dynamics of strongly-coupled counterions still retain diffusive behaviors within each Wigner–Seitz cell (i.e., $l_{c}∼a$), which is physically plausible (see also Figure 3).

## 5. Summary and Conclusions

We have investigated stochastic density fluctuations $n=\rho -\rho_{\infty }$ around the ground state distribution ($\rho_{\infty }\propto e^{-z_{0}/\lambda }$) of strongly coupled counterions near a single charged plate, focusing especially on the transverse dynamics parallel to the charged plate at $z_{0}=0$. The key to treating the stochastic dynamics is to use the DK equation of overdamped Brownian particles [13,14,15,16,17,18,19,20,21,22,23,24,25,26,27] that is linearized by expanding the first derivative of a free energy functional of given density, (δF[ρ]/δρ) around the ground state density $\rho_{\infty }$. As a result, we have obtained the linear DK Equation (Equation 37), which is applicable to the longitudinal and transverse dynamics of counterions in the strong coupling regime where the stationary density distribution has been investigated using Monte Carlo simulations [8,9,10,11].

The linear DK equation allows us to quantitatively investigate the dynamical crossover of transverse density fluctuations along the plate surface. Accordingly, we have found a crossover scale, given by Equations (Equation 46) and (Equation 47), above which the transverse density dynamics appear frozen, generating white noise that is uncorrelated with respect to time and space. Below the crossover scale, on the other hand, diffusive behavior of counterions can be observed along the plate surface, as illustrated in Figure 3. For instance, the crossover length $l_{c}$ is of the order of the Wigner–Seitz radius *a* when $\Xi ∼10^{3}$. Furthermore, the longitudinal dynamics vertical to the plate arises from the gradient of a fluctuating density field along the *z*-axis, producing additional contributions to the transverse dynamics, such as electrical reverse-flow as well as advective flow (see Figure 2). The electrical reverse-flow would be crucial in experimental situations where mobile ions, including not only counterions but also added salt, are affected considerably by the longitudinal dynamics. This remains to be addressed in a quantitative manner, by extending the present formulation to multi-component systems. 

## Figures and Tables

**Figure 1 entropy-22-00034-f001:**
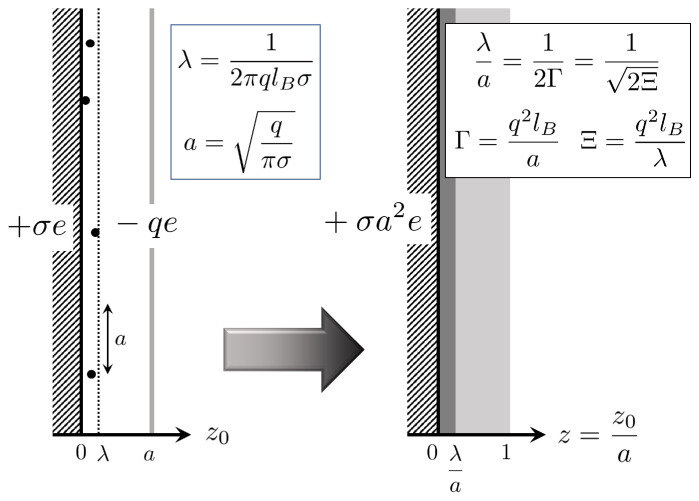
Two schematics illustrating the side view of one charged plate system that consists of a positively charged plate carrying a surface charge density +σe and negatively charged counterions with valence *q*. Our scaling ($z=z_{0}/a$) of an actual system depicted on the left side implies a coarse-grained system on the right hand side, where *a* denotes a mean separation between counterions, provided that all of the counterions are condensed on the oppositely charged plate uniformly: $\pi a^{2}\sigma =q$. The Gouy–Chapman length $\lambda \equiv 1/(2\pi ql_{B}\sigma )$ is also indicated. This paper adopts the coupling constant Γ, defined by $\Gamma =q^{2}l_{B}/a$ that applies to the 2D one-component plasma (OCP), instead of the conventional one, $\Xi =q^{2}l_{B}/\lambda$, used for the counterion system.

**Figure 2 entropy-22-00034-f002:**
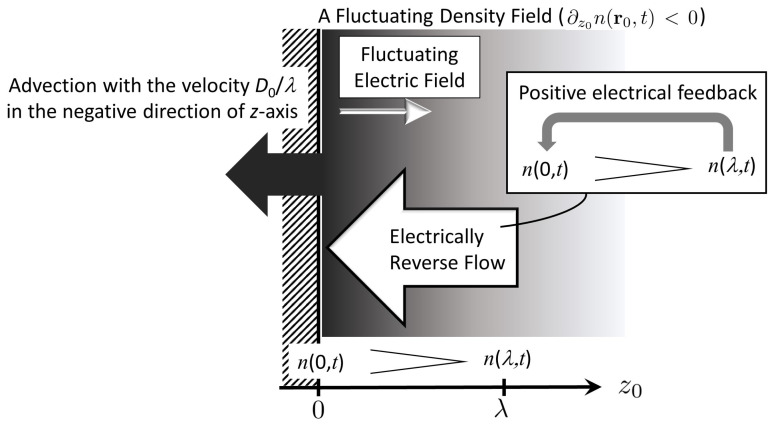
A schematic of anisotropic fluctuations due to longitudinal terms underlined in Equation (37). Here we consider the case that the fluctuating density $n(r_{0},t)$ decreases with $z_{0}$, and n(0.t) and n(λ,t) are abbreviations of $n\left(r_{0},t\right){|}_{z_{0}=0}$ and $n\left(r_{0},t\right){|}_{z_{0}=\lambda }$, respectively. The advective flow, or migration of density fluctuations as a whole, is always in the negative direction along the $z_{0}$-axis. Meanwhile, the electrical reverse-flow is also in the negative direction, irrespective of the sign of $\partial_{z_{0}}n$, or $E_{z}$. Accordingly, the latter flow acts as a positive feedback of density fluctuations for $\partial_{z_{0}}n<0$.

**Figure 3 entropy-22-00034-f003:**
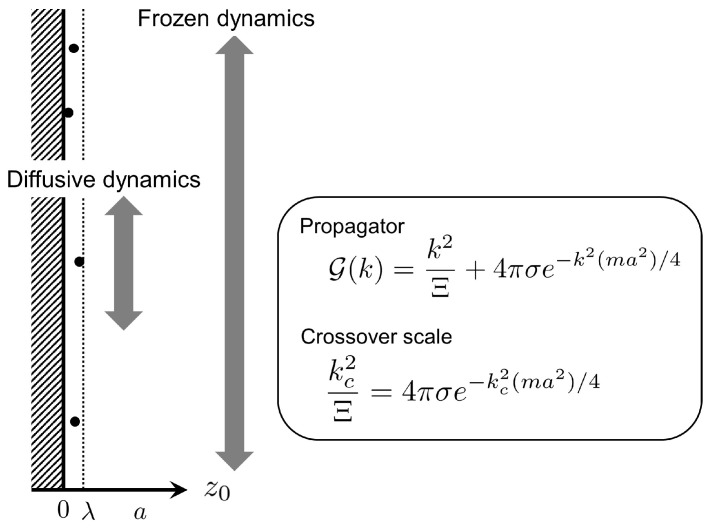
A schematic of dynamical crossover, showing that diffusive dynamics of strongly coupled counterions along the plate surface can be observed within the scale of Wigner–Seitz cell. The determining equation for the crossover scale $k_{c}$ is also given, based on the expression of propagator G(k) given by Equation (39).

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
