# Peer review of "Transverse Density Fluctuations around the Ground State Distribution of Counterions near One Charged Plate: Stochastic Density Functional View"

_entropy, 2019, doi:10.3390/e22010034_

Round 1

Reviewer 1 Report

The author explored the stochastic density fluctuations around the ground state distribution of strongly coupled counterions nearby a charged plate. In doing so, the author examined the dynamic of overdamped Brownian particle perpendicular
to a charged plate, or equivalently, the linear version of the Dean-Kawasaki equation for the stochastic dynamics is employed.  Interestingly, it was shown the existence of a cross-over scale (Wigner-Seitz radius) which separates the frozen state of the counterions from the diffusive regime. All in all, the paper is well-written and nicely organized so I think it should be published in its current form.

Reviewer 2 Report

This paper derives a linearized Dean-Kawasaki equation, showing it to be equivalent to a diffusion equation under a potential of mean force.  The screened Coulomb form of the direct correlation function observed in plasmas is then adopted to describe the limiting behavior of ionic motions.

The technical results are clearly derived, with two key reservations.  First, the assumption (after Eq. 19) that c/rho_inf << 1 does not seem to be justified in a plasma.  This is especially important for the main result, since the approximation assumes the ideal gas limit, and the fluctuations under strong coupling are then found to follow ideal gas behavior.  A Monte Carlo calculation of the time-dependence would be ideal.

The description of forward and reverse flows in Fig. 2 (and in the text following Eq. 35) could be made much more rigorous by substituting the linear or exponential density profiles and computing magnitudes for each flux.

The introduction spends too much effort describing types of ions.  The paper would be better presented by starting with lines 88-90 (starting with "The aim of this paper is to investigate anisotropic fluctuations...").

Replacing the preceding paragraphs with a single one listing physical examples, like DNA, lysozyme or silica nanoparticles rather than by macroion shapes.  Also, for lines 25-26, 0.1 nm ions are monatomic, and polyatomic ions have internal structure.

Other specific comments are below:

In Eq. 10, F[rho] depends on J. How is J determined by the left side?

A reference should be cited for Eq. 16, since it seems (16) may satisfy (15), but not be unique.

Eq. 17 should define n(r).

Reviewer 3 Report

In the manuscript, the author addresses a problem inspired from electrostatics, and studies the distribution of counterions near a charged surface. He calculates analytically the anisotropic fluctuations of the counterions, employing the stochastic density functional equation, itself based on the Dean-Kawasaki equation.

The paper is well-written and the results are presented clearly, with a few sketches presenting the main physical effects. I appreciate that some technical points are developed in the appendices, thus making the main text more readable.

However, it is not easy to evaluate the impact of the results, since they rely on different approximations, whose range of validity is not always specified. For instance, the results derived in Section 3.1 rely on some approximation which is made explicit in Appendix D. Section 4 involves an approximate form of the propagator G. The free energy F is evaluated in the mean-field limit. Do these approximations originate from the same hypotheses? Are they consistent one with another? Were they confronted to numerical simulations, or justified in other references?

I think the readers would appreciate a deeper insight in these approximations and the hypotheses on which they rely. A clear distinction between exact and approximate results, as well as a discussion on the range of validity of these approximations and their relation to realistic experimental situations would be appreciated.

I also have a few remarks about the introduction of the manuscript:
- It could be easier to read with a sketch of the generic situation described by this theoretical framework.
- When the stochastic DFT is introduced (lines 88 to 108) it is not clear what 'linear' refers to: what is the theory usually linearised with respect to? What is typically the small parameter: is it the density fluctuations, the interaction strength?...

In summary, I am willing to support publication in Entropy if the author addresses the points raised in this report.

Round 2

Reviewer 3 Report

The author made a substantial effort to improve the clarity and presentation of the results, therefore I am now willing to recommend publication in Entropy.